# Application of a Fluorescent Biosensor in Determining the Binding of 5-HT to Calmodulin

**L. X. Vásquez-Bochm** [1], **Isabel Velázquez-López** [2], **Rachel Mata** [1], **Alejandro Sosa-Peinado** [2], **Patricia Cano-Sánchez** [3] **and Martin González-Andrade** [2,*]

1   Facultad de Química, Universidad Nacional Autónoma de México, Ciudad de México 04510, Mexico; luz@bq.unam.mx (L.X.V.-B.); rachel@unam.mx (R.M.)
2   Laboratorio de Biosensores y Modelaje Molecular, Departamento de Bioquímica, Facultad de Medicina, Universidad Nacional Autónoma de México, Ciudad de México 04510, Mexico; isavel@bq.unam.mx (I.V.-L.); asosa@bq.unam.mx (A.S.-P.)
3   Instituto de Química, Universidad Nacional Autónoma de México, Ciudad de México 04510, Mexico; pcano@iquimica.unam.mx
*   Correspondence: martin@bq.unam.mx

**Abstract:** Here, we show the utility of the fluorescent biosensor *h*CaM-M124C-*mBBr* in detecting and determining the affinity of serotonin (5-HT). We obtained a $K_d$ of 5-HT (0.71 μm) for the first time, the same order of magnitude as most anti-CaM drugs. This data can contribute to understanding the direct and indirect modulation of CaM on its binding proteins when the 5-HT concentration varies in different tissues or explain some of the side effects of anti-CaM drugs. On the other hand, molecular modeling tools help the rational design of biosensors and adequately complement the experimental results. For example, the docking study indicates that 5-HT binds at the same site as chlorpromazine (site 1) with a theoretical $K_i$ of 2.84 μM; while the molecular dynamics simulations indicate a stability of the CaM–5-HT complex with a theoretical $\Delta G$ of $-4.85$ kcal mol$^{-1}$, where the enthalpy contribution is greater. Thus, the combination of biotechnology and bioinformatics helps in the design and construction of more robust biosensors.

**Keywords:** biosensor; calmodulin; 5-HT; docking; molecular dynamic simulation

## 1. Introduction

Biosensors present a hybrid mechanism to transform information from chemical interactions into analyzable signals through biochemical mechanisms. These systems are generally a receptor system, a biological component that specifically interacts with an analyte and transduces the signal with a detector system. The transductor component has been the focus of much attention devoted by chemists and biologists over the past decades to develop "biosensors" that allow the tracking or detecting of a small molecule of interest in a minimal amount of time. Especially, fluorescent biosensors are currently the most widely used due to their high sensitivity and selectivity, sufficient temporal and spatial resolution, and low cost of use [1]. The advantage of these systems is that the biological component gives it high selectivity, and the transducer component provides high sensitivity and reproducibility [2].

A molecular target drug that interacts with molecules and regulates many metabolic pathways is the protein calmodulin (CaM). CaM has been the subject of various studies of computer, thermodynamic, structural, evolutionary, and pharmacological types [3–9]. This protein is one of the most abundant, ubiquitous, and conserved: more than 60% of it is conserved among eukaryotes and 100% conserved among vertebrates [10]. The sequence of CaM comprises 148 amino acids formed by two domains containing each domain two Ca$^{2+}$-binding loops known as EF-hands. These domains are separated by a long central helix giving this protein a dumbbell shape. CaM has no enzymatic activity

but plays an essential role in calcium signaling pathways. CaM interacts with many proteins to activate or regulate intracellular calcium concentration [11,12]. This protein is a molecular target of compounds with pharmacological activity, such as anti-cancer, antipsychotic, antidepressant, muscle relaxant, and local anesthetic drugs. Moreover, it involves physiological processes such as muscle contraction, fertilization, cell proliferation, vesicular fusion, apoptosis, and others [13–17].

Figure 1 shows the development of the *h*CaM-M124C-*mBBr* biosensor. The development of this biosensor consisted of a rational design to identify the best site to mark the protein with a fluorophore to monitor both local microenvironment changes and interactions of potential CaM protein inhibitors. A characteristic of the CaM protein is that it does not present cysteine residues in its wild form, so site-specific labeling with thio-reactive fluorophores such as monobromobimane fluorophore (*mBBr*) can be carried out with the mutation of any amino acid by a cysteine. This development shows high sensitivity ($\Phi = 0.49$), and thanks to its extrinsic fluorescence, it does not interfere with most anti-CaM and endogenous molecules ($\lambda_{exi} = 381$ and $\lambda_{emi} = 473$ nm) [18].

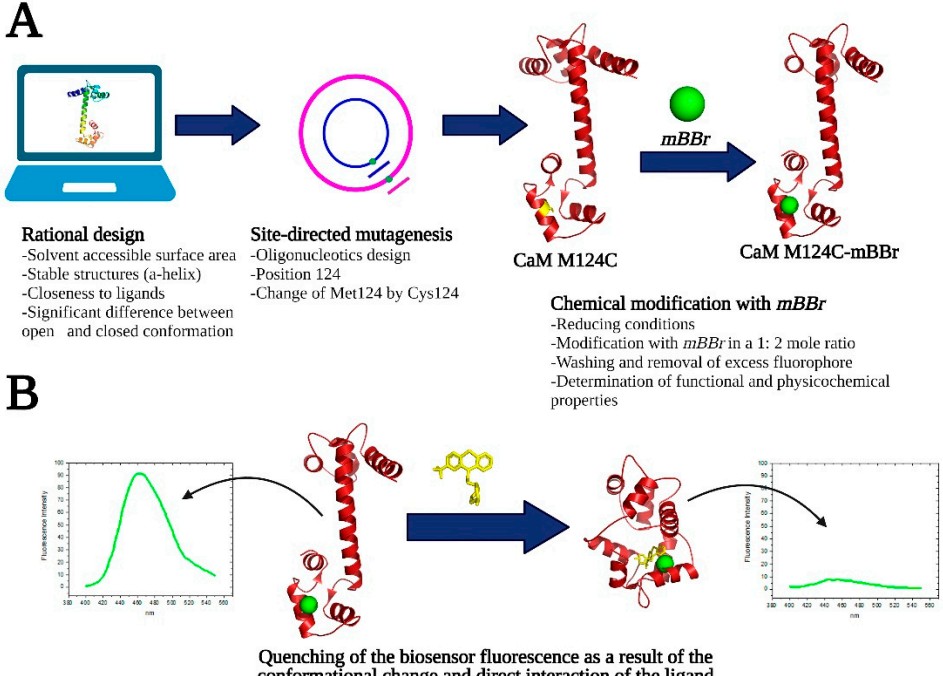

**Figure 1.** Development and functionality of the *h*CaM-M124C-*mBBr* biosensor. Panel (**A**) shows the complete development for the construction of the fluoresce biosensor, highlighting the critical points of each step of the process. Panel (**B**) presents the functionality of this development, where the change of the microenvironment and direct interaction of the ligand turns off the fluorophore.

Some drugs of everyday use such as chlorpromazine (CPZ), trifluoperazine (TFP), fluoxetine (FLU), and vinblastine (VBT) [19–24] are linked directly or indirectly to CaM and its related proteins, and some endogenous molecules such as serotonin (5-HT) and dopamine (DOP) could also bind to CaM and affect the potency of the aforementioned drugs. 5-HT is a molecule that acts as a biochemical messenger and regulator. This metabolite can be found mainly in the gastrointestinal tract, central nervous system, and platelets. Some important physiological functions in which 5-HT participates are gastrointestinal motility, neuronal communication, cardiovascular integrity, and hemostasis. Multiple receptor families explain this biochemical mediator's broad physiological actions and distribution [25–27]. DOP is an endogenous molecule belonging to the catecholamines family, which plays important roles at neuronal and physiological levels. In the CNS,

DOP behaves like a neurotransmitter, and in the rest of the body, it plays a diversity of roles [28,29].

On the other hand, computational tools such as docking and molecular dynamics simulations are ideal for obtaining theoretical thermodynamic and union parameters, compared with experimental data, and help propose structural models of protein–ligand complexes [30–32]. In this work, we show the usefulness of the *h*CaM-M124C-*mBBr* biosensor in detecting and determining the affinity of 5-HT for the CaM protein for the first time. We show that 5-HT binds CaM with a $K_d$ of 0.71 μM and is of great importance. It can be related to the side effects of some CaM agonists used as antidepressants (FLU, CPZ, and TFP) and chemotherapeutic agents (VBT). Docking and molecular dynamics studies correlate adequately with experimental data and provide structural details at the atomic level of the CaM–ligand interaction.

## 2. Materials and Methods

### 2.1. Materials

The biosensor used in this work (*h*CaM M124C-*mBBr*) was produced according to the methodology described by González-Andrade, M., et al. [18]. CPZ, 5-HT, DOP, and AMA were obtained from Sigma-Aldrich Química (Toluca, México). All other consumables were analytical grade.

### 2.2. Steady-State Fluorescence

All measurements were conducted with an ISS–PC1 spectrofluorometer (ISS, Champaign, IL, USA) with sample stirring at 37 °C. The *h*CaM M124C-*mBBr* (1 μM) was incubated in buffer (10 mM of potassium acetate pH 5.1 and 10 μM of CaCl$_2$). The slits used in the acquisition of the fluorescence spectra had a width of 4 and 8 nm for the excitation and emission, respectively. The $\lambda_{exi}$ was 381 nm, and the $\lambda_{emi}$ was collected from 415 to 550 nm. The fractional degree of saturation biosensor–ligand complex (*y*) was estimated by charges in fluorescence on ligand corresponding to $y = (F - F_0)/(F_\infty - F_0)$, where $F_\infty$ represents the fluorescence intensity at saturation of the ligand, *y* is graphed as a function of the protein/ligand relation (*L*), and the apparent dissociation constants ($K_d$) and stoichiometric (*S*) were estimated by fitting to Equation (1):

$$y = \frac{(1 + K_d/S + L/S) - \sqrt{(1 + K_d/S + L/S)^2 - 4L/S}}{2} \tag{1}$$

where *y* is the fractional degree of fluorescence intensity, $K_d$ is the apparent dissociation constant, *L* is the protein/ligand relation, and *S* is stoichiometric. OriginPro version 9.0 64-bit SR2 (OriginLab, Northampton, MA, USA) was used to process all data.

### 2.3. Obtaining and Preparing the Files of the Three-Dimensional Structures

The three-dimensional file structures corresponding to the CaM protein were obtained from the Protein Data Bank (PDB, http://www.rcsb.org, accessed on 30 April 2021). The CaM–ligand complexes, the X-ray structure of CaM with calcium, and ligand TFP named 1A29.pdb (1A29, close form of the CaM) refined at 2.7 Å were chosen [33]. The ligands were obtained from the PDB co-crystillized structure, and when the crystals were not available, their structures were constructed using AVOGRADRO software (version 2, Free Software Foundation, Boston, USA) [34]. Subsequently, the constructed ligands were minimized using Gaussian 09, revision A.02 (Gaussian Inc., Wallingford, CT, USA) at DFT B3LYP/3-21G level of theory. For the ligands used in the molecular dynamics simulations, these were parameterized using the Antechamber program in AmberTools [35].

### 2.4. Docking

The docking studies were carried out using the three-dimensional structure of CaM obtained by X-ray crystallography (1A29.pdb). Using the idealization application of

Rosetta 3.1 release, the three-dimensional protein structure was reconstructed and refined prior to the docking studies [36]. AutoDockTools 1.5.4 was used to prepare CaM and its ligands. The preparation of CaM consisted of adding polar hydrogen atoms, Kollman united-atom partial charges, and for the ligands computing Gasteiger–Marsili formalism charges, rotatable groups which were assigned automatically, as were the active torsions. The docking was run using AutoGrid4 and AutoDock4 version 4.2 software [37], initially in a 60 Å × 60 Å × 60 Å grid box and to refine the best pose in a 30 Å × 30 Å × 30 Å grid box using the recommended parameters. The analysis of the docking was made with AutoDockTools using cluster analysis and the program PyMOL [38].

*2.5. Molecular Dynamics Simulation*

The best poses of the ligands obtained from the docking studies were parameterized with Antechamber (a set of auxiliary programs for molecular mechanic studies) to build the CaM–ligand complexes used in the molecular dynamics simulation studies. The *LEaP* module from AMBER was used to assemble the initial topology and coordinate files [35,39]. Each file was subjected to the subsequent protocol: hydrogens were added, ions counterions were added to neutralize the system, the system was solvated in an octahedral box of solvent explicit (TIP3P model water molecules) localizing the limits at 12 Å from the complex surface. Molecular dynamics simulations were made at 1 atm and 298 K, preserved with the Berendsen barostat and thermostat, using periodic boundary conditions and particle mesh Ewald sums (grid spacing of 1 Å) for treating long-range electrostatic interactions with a 10 Å cutoff for computing direct interactions. The SHAKE algorithm was used to satisfy bond constraints, allowing employment of a 2 fs time step for the integration of Newton's equations [40,41]. Amber f19SB force field [39,42] parameters were used for protein and Gaff2 force field for the ligands [43]. All calculations were run using a GPU-accelerated molecular dynamics simulation engine in AMBER (pmemd.cuda) [44]. The protocol consisted in performing a minimization of the initial system, followed by 50 ps heating step at 298 K, 50 ps for equilibration at constant volume, and 500 ps for equilibration at constant pressure. Several independent 100 ns molecular dynamics simulations were performed. Structures were saved at 100 ps intervals for subsequent analysis, using the CPPTRAJ program [45].

*2.6. Binding Free Energies Calculated by Molecular Mechanics/Poisson–Boltzmann Surface Area (MM/PBSA)*

The calculation of the binding free energies consists of the combination of molecular mechanical energy with implicit solvation models. In MM/PBSA, binding free energy ($\Delta G_{bind}$) between a protein and a ligand to form a protein–ligand complex is calculated as:

$$\Delta G_{bind} = \Delta H - T\Delta S \approx \Delta E_{MM} + \Delta G_{Sol} - T\Delta S \tag{2}$$

$$\Delta E_{MM} = \Delta E_{Internal} + \Delta E_{Electrostatic} + \Delta E_{Vdw} \tag{3}$$

$$\Delta G_{Sol} = \Delta G_{PB} + \Delta G_{SA} \tag{4}$$

where $\Delta E_{MM}$, $\Delta G_{Sol}$, and $T\Delta S$ are the changes of the gas phase molecular mechanics energy, the solvation free energy, and the conformational entropy upon binding, respectively. $\Delta E_{MM}$ comprises $\Delta E_{Internal}$ (bond, angle, and dihedral energies), $\Delta E_{Electrostatic}$ (electrostatic energies), and $\Delta E_{Vdw}$ (Van der Waals energies). $\Delta G_{Solv}$ is the sum of electrostatic solvation energy (polar contribution) $\Delta G_{PB}$ and non-electrostatic solvation component (non-polar contribution) $\Delta G_{SA}$. The polar contribution is calculated using the Poisson–Boltzmann surface area model, while the non-polar energy is estimated from the solvent accessible surface area (*SASA*). The conformational entropy change (*T*$\Delta S$) was computed by normal mode analysis from a set of conformational snapshots taken from the molecular dynamics simulation [46]. All calculations were made using a system HP Cluster Platform 3000SL (Hewlett-Packard Development Company, Spring, TX, USA), supercomputer "MIZTLI" with a processing capacity of 118 TFlop/s. It had 5312 Intel E5-2670 processing cores,

16 NVIDIA m2090 cards, a total RAM of 15,000 GB and a mass storage system of 750 TB (http://www.super.unam.mx/, accessed on date 30 April 2021).

## 3. Results and Discussion

### 3.1. Analysis of the CaM Complex with Clorpromazine (1:4) for Biosensor Design

It has been reported that CaM can bind from one to four ligands depending on its size; for example, CaM–KAR-2 (1:1), CaM–AAA (1:2), and CaM–TFP (1:4) complexes [5,47]. Figure 2 shows an analysis of the structure of the CaM–CPZ complex. Site 1 is formed by residues Phe92, Ile100, Leu105, Met124, Phe141, and Met144; site 2 (Glu11, Glu14, Ala15, Leu18, Phe19, Val35, Leu39, Met72, Met109, Leu112, and Glu114); site 3 (Met36, Leu39, Glu41, Glu84, Ala88, Val91, and Phe92), and site 4 (Phe19, Ile27, Leu32, Met51, Ile52, Val55, Phe68, and Met71). Although the affinity of the binding sites can vary depending on the ligand, the residues corresponding to site 1 have been reported as constant in most co-crystallized structures and molecular dynamics and docking studies [20,22,48–52]. Additionally, this site is also involved in interactions with CaM-binding peptides and proteins [52,53]. The Met124 residue is part of site 1, so covalently labeling the protein with the *mBBr* fluorophore was essential for the *hCaM-M124C-mBBr* biosensor to have the ability to monitor the binding of most of the ligands, peptides, and proteins.

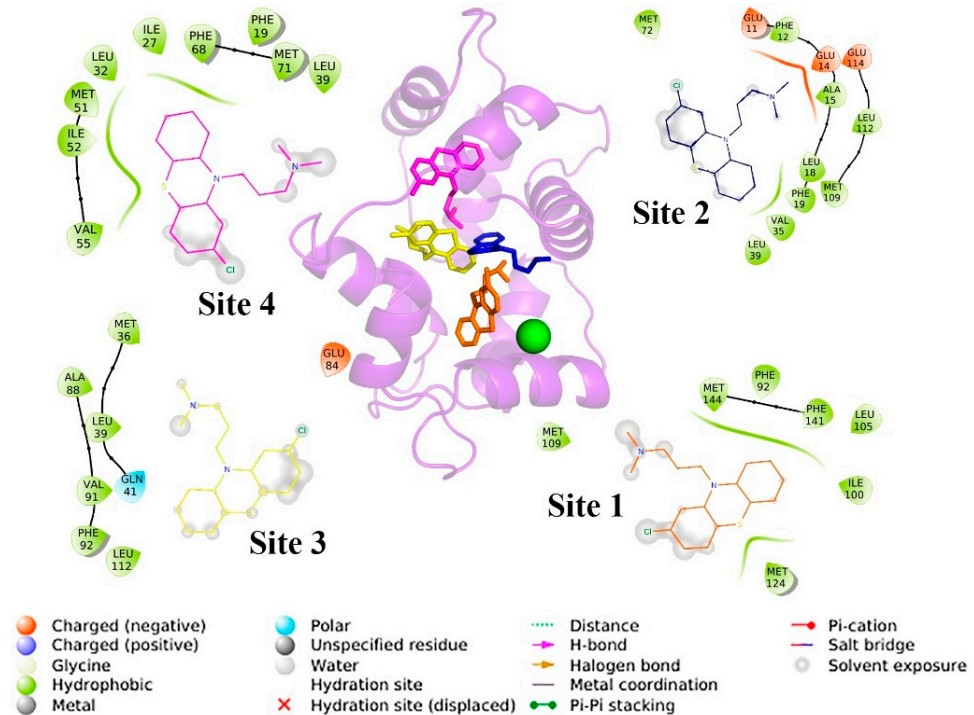

**Figure 2.** Analysis of the CaM-CPZ complex (1:4). In the center, CaM is shown in purple cartoons with 4 CPZs (CPZ-1; orange stickers), (CPZ-2; blue stickers), (CPZ-3; yellow stickers), and (CPZ-4; purple stickers); and in the green sphere, position 124 is where the fluorophore of the biosensor is attached. In the periphery, the analysis of the interactions with the residues is at 4 Å.

### 3.2. Purification and Chemical Modification with mBBr of the CaM M124C Protein

The *h*CaM M124C recombinant protein was purified by hydrophobic exchange chromatography on a Phenyl-Sepharose CL-4B column (Figure S1). Subsequently, it was chemically modified with *mBBr* fluorophore to obtain *h*CaM M124C-*mBBr*, which was repurified using molecular exclusion chromatography on a superdex75 column and monitored at two wavelengths, 276 and 381 nm, corresponding to the aromatic residues of CaM and *mBBr*, respectively (Figure S2). Figure 3A shows the follow-up of obtaining the *h*CaM M124C-*mBBr* biosensor. The lanes of the acrylamide gel were analyzed with the Image J

program (http://rsb.info.nih.gov/ij/, accessed on 30 April 2021), obtaining the intensity of each lane according to the tonality scale of the digitized and computerized image. The amount of recombinant protein was estimated to be around 80% of the total protein, with a purity of 98%, with a yield of around 160 mg of *h*CaM M124C-*mBBr* biosensor per liter of medium. The spectroscopic properties of the CaM M124C mutant and the biosensor are shown in Figure 3B, where we observe an absorption maximum at 381 and 276 nm for *h*CaM M124C-*mBBr* and *h*CaM M124C, respectively.

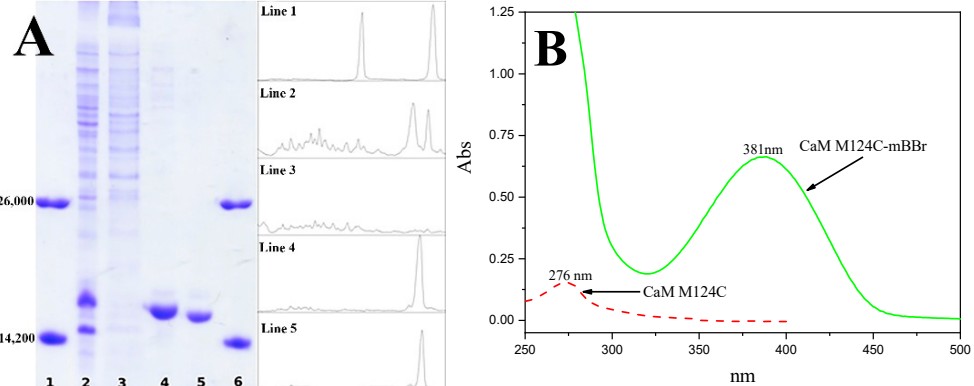

**Figure 3.** (**A**) Follow-up of the purification of the *h*CaM M124C protein by SDS-PAGE. Lane 1 and 6 a-Lactoalbumin (14,200 Da) and LAO protein (26,000 Da), lane 2 supernatant after sonication, lane 3 fraction without binding to the Phenyl-Sepharose CL-4B column, lane 4 fraction corresponding to the *h*CaM M124C from the Phenyl-Sepharose CL-4B column and lane 5 fraction of the *h*CaM M124C-*mBBr* after passing through the size exclusion column (Superdex75). (**B**) The absorption spectra of the *h*CaM M124C (-) and *h*CaM M124C-*mBBr* (—).

### 3.3. Determination of the Binding Affinity of 5-HT and CPZ with the Biosensor hCaM M124C-mBBr

To determine the affinity of 5-HT using the biosensor *h*CaM M124C-*mBBr* in calcium-saturating conditions (10 µM), these were titrated by adding increasing amounts of 5-HT to determine the $K_d$. Figure 4 shows fluorescence spectra of the biosensor with different concentrations of ligands; the differences observed in the fluorescence signal were used to calculate the $K_d$, using a one-site binding model equation (Equation (1), see method section). Table 1 shows the results obtained from the fluorescence titrations of the compounds. CPZ has been previously reported to bind to a CaM–EGFP fusion protein in the presence of $Ca^{2+}$, but this system does not respond to 5-HT [54]. In the case of 5-HT, previous studies using CaM labeled at position 109 did not observe the response of this biosensor either [55]. However, we can perform a complete titration with 5-HT and obtain a $K_d$ of 0.71 µM, using the *h*CaM-M124C-*mBBr* biosensor (Figure 4A), which is within the range expected for most organic molecules. Another neurotransmitter, DOP, was analyzed with our biosensor and showed no response (Figure S3A), agreeing with both CaM–EGFP and the CaM labeled at position 109. CaM has specific binding sites for inhibitors and proteins, so inhibitors share standard structural features. In general, the inhibitors of CaM are structures of resonant type, with a zone of character highly hydrophobic and an electronegative pole. The structural differences between 5-HT and DOP comprise those primarily in the conjugated system. The DOP molecule consists of a catechol structure, while 5-HT is synthesized from the essential amino acid L-Tryptophan, so its conjugate system is an indole. Amantadine (AMA) was used as a negative control; this antiviral drug structurally does not exhibit the features described above. Figure S3B shows that the addition of about 80 µM does not change the fluorescence signal of the biosensor.

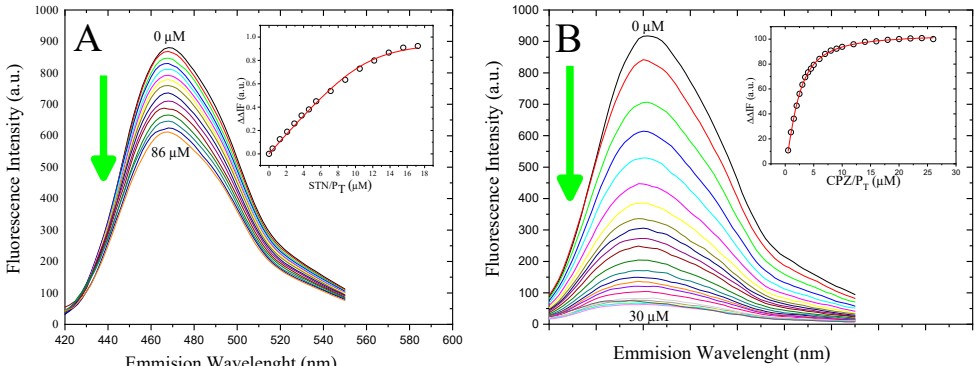

**Figure 4.** Fluorescence spectra and titration curves of *h*CaM M124C-*mBBr* with 5-HT (**A**) and CPZ (**B**). Buffer was 10 mM of potassium acetate pH 5.1 at 37 °C. The absolute changes of maximal fluorescence emission were corrected for light scattering effects and plotted against the ligands-to-total-protein ratio (insets). The continuous line in the insets comes from the fitting of data to the binding model (Equation ((1)) to obtain the $K_d$.

**Table 1.** Experimental and theoretical binding properties of CaM–ligand complexes.

| Complex | Experimental Studies | | Docking Studies | | MD Studies | | |
|---|---|---|---|---|---|---|---|
| | $\Delta G_{exp}$ (kcal mol$^{-1}$) | $K_d$ [1] (M) | EFEB [2] (kcal mol$^{-1}$) | $K_d$ [3] (M) | $\Delta H$ (kcal mol$^{-1}$) | $T\Delta S$ (kcal mol$^{-1}$) | $\Delta G_{cal}$ (kcal mol$^{-1}$) |
| CaM–CPZ | −8.19 | $0.97 \times 10^{-6}$ | −8.24 | $0.90 \times 10^{-6}$ | −22.27 ± 2.5 | −17.75 ± 7.7 | −4.52 ± 1.8 |
| CaM–5-HT | −8.38 | $0.71 \times 10^{-6}$ | −7.56 | $2.84 \times 10^{-6}$ | −21.38 ± 2.7 | −16.52 ± 3.4 | −4.85 ± 0.4 |
| CaM–DOP | - | - | −7.21 | $2.96 \times 10^{-6}$ | −11.85 ± 3.5 | −13.60 ± 5.24 | +1.74 ± 0.8 |
| CaM–AMA | - | - | −7.74 | $2.13 \times 10^{-6}$ | +133.30 ± 13.8 | +15.60 ± 5.16 | +117.69 ± 21.2 |

[1] Apparent dissociation constants at 298.15 K; [2] Estimate Free Energy of Binding; [3] Theory inhibitor constants at 298.15 K.

### 3.4. Relevance of Interaction between CaM–5-HT

The experimental determination of the binding of 5-HT to CaM is important since CaM interacts with a wide range of membranal receptors such as epidermal growth factor receptor, the cytoplasmic domain of platelet glycoprotein VI, and some G protein-coupled receptors (GPCRs) [56,57]. The GPCRs that interact with CaM are glutamate subtype 5, D2-dopamine, m-opioid, V$_2$-vasopressin, and 5-HT$_{1A}$ receptors [58–62]. Thus, CaM interactions with these receptors play an essential role in modulating and signaling different metabolic pathways associated with these GPCRs.

5-HT, being an endogenous metabolite, can vary the concentration in different conditions in the body. Therefore, if it forms a CaM–5-HT complex in addition to performing its function, it could indirectly regulate some receptors modulated by CaM, including the 5-HT receptor itself. Another possibility is that 5-HT can compete with CaM agonist drugs, synergizing or abolishing activity of the latter.

Figure 5 outlines a possible scenario of the implications of serotonin being able to bind and inactivate CaM. CaM in the presence of calcium exposes hydrophobic patches capable of interacting with various proteins, such as the 5-HT$_{1A}$ receptor. It has been described that it possesses two CaM-binding regions (residues 217–237 and 331–349), which modulate the activity of the 5-HT receptor-protein G complex.

Using the *h*CaM-M124C-*mBBr* biosensor, we demonstrate that 5-HT binds and inactivates to CaM, since it stabilizes the closed conformation (inactive). Therefore, 5-HT may play an important role in modulating many GPCRs through the inactivation of CaM.

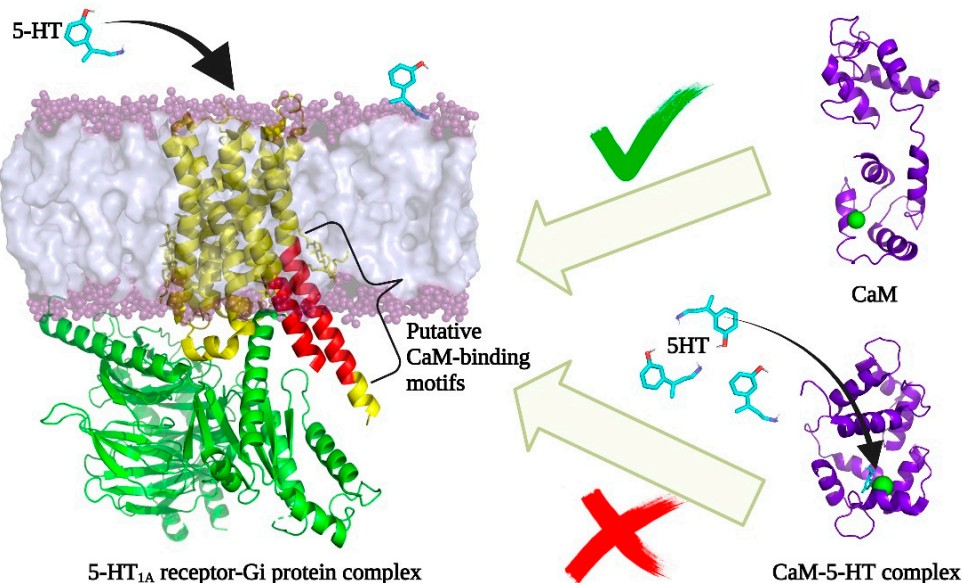

**Figure 5.** Scheme of a possible indirect modulation of 5-TH on the serotonin-bound serotonin 1A (5-HT$_{1A}$) receptor through CaM. On the left is shown 5-HT$_{1A}$ receptor-Gi protein complex; 5-HT$_{1A}$ receptor in yellow cartoons (red cartoons represent CaM interaction sites), and Gi protein in green cartoons, and the lipid bilayer in purple surface. CaM and CaM–5-HT complex are shown in purple cartoons and 5-HT in cyan stickers. The structural model of the 5-HT$_{1A}$ receptor-Gi protein complex was obtained from the PDB code 7E2Y.pdb, for the CaM and CaM–5-HT complex 1Y0V.pdb and 1A29.pdb, respectively.

### 3.5. Docking

Docking studies show that 5-HT binds to site 1, interacting directly with position 124 and forming a hydrogen bridge, where the fluorophore is bound in our biosensor so that this interaction can explain the quenching of fluorescence (Figure 6). The $K_i$ for 5-HT was 2.84 μM; for DOP and AMA the $K_i$ values were 2.96 and 2.13 μM, respectively (Table 1). The data of DOP and AMA should be taken with caution since these compounds do not respond to interacting with *h*CaM-M124C-*mBBr*. Therefore, docking studies can present false-positive results when there is not enough experimental information. However, when experimental information is available, it is a powerful tool for studying protein–ligand interactions at the atomic level and making good correlations. On the other hand, the comparison of the $K_i$ with the experimental $K_d$ of 5-HT and CPZ are good since they are in the same range of μM, and for 5-HT there is a difference of 3 times, and for CPZ they are practically the same. The best pose of the 5-HT obtained from the docking was used to perform the molecular dynamics simulations, which is of great importance since there is no structural model of the complex to be studied in our case, the CaM–5-HT complex.

### 3.6. Molecular Dynamics Simulations

Unlike docking studies, molecular dynamics simulations represent the movement of all the atoms of a system, where they vibrate and move in a defined time [63]. A key element for conducting molecular dynamics simulation studies is the starting structure of the CaM–ligand complex. The initial structures were obtained from docking studies and available experimental information of CPZ. Another critical point is the optimal time to perform a molecular dynamics simulation, depending on the studied particle system. Figure S4 shows the changes in system energy during the simulation process of the CaM–CPZ complex, which gives us the guidelines to establish the simulation times for this system. We calculated some thermodynamic parameters of all complexes using the method normal mode analysis to identify the enthalpy and entropy contributions of binding ligands to CaM. This calculation estimates a $\Delta G_{cal}$ and allows determining the

enthalpy and entropy contributions associated with the protein–ligand interaction directly related to molecular recognition. Table 1 shows the theoretical thermodynamic parameters of the molecular dynamics simulation trajectories.

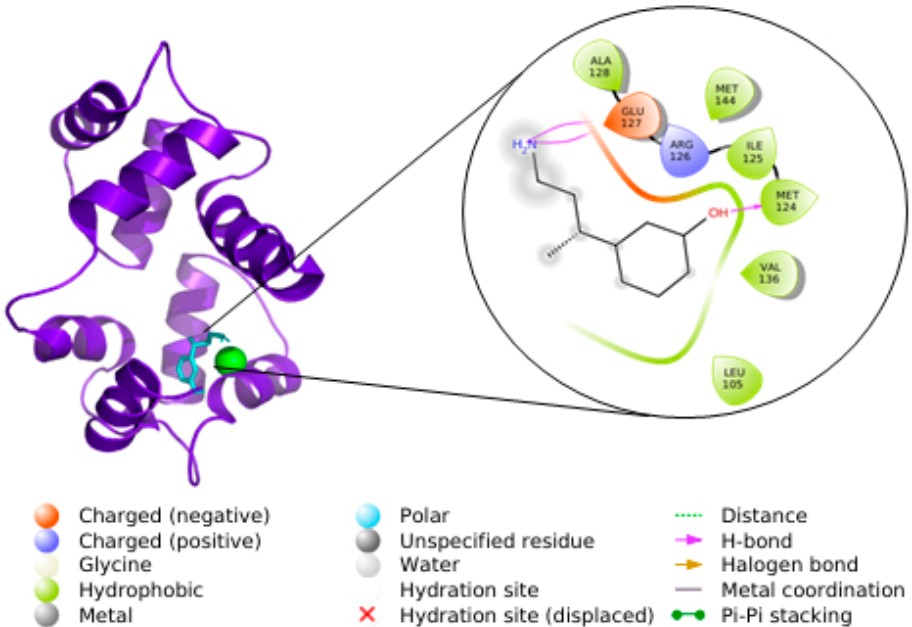

**Figure 6.** Docking of CaM–5-HT complex. CaM is shown in purple cartoons, position 124 in green spheres, and 5-HT in cyan stickers. The analysis of the interaction residues at 4 Å of 5-HT is shown in a zoom. The figures were made with PyMOL and Maestro.

The energetic compound in the docking results (EFEB) shows a good correlation with experimental results for CaM–5-HT and CPZ complexes. We estimate interaction energy for the CaM–DOP and CaM–AMA complexes even though we did not observe a union experimentally. However, molecular dynamics studies calculate a negative $\Delta G_{cal}$ for CaM–CPZ and CaM–5-TH complexes (−4.52 and −4.85, respectively) and a positive $\Delta G_{cal}$ for CaM–DOP and CaM–AMA complexes (+1.74 and +117.69, respectively) which is consistent with the experimental data. For CaM–5-HT and CaM–CPZ complexes, the enthalpy component is the one that contributes most to $\Delta G$, which may be the result of the direct interaction of the ligands with CaM.

Figure 7A shows the root mean square deviations (RMSDs) vs. Time for 100 ns molecular dynamics simulation; RMSDs show slight fluctuation (1–3 Å) over time for CaM–CPZ and CaM–5-HT complexes, and for CaM and the CaM–DOP complex (~10 Å). The large RMSD observed for CaM can be attributed to the dynamics of the protein itself, which fluctuates between "closed" and "open" forms when a ligand binding stabilizes a conformation; in this case, closed conformation. For the CaM–DOP complex, DOP does not bind to the protein, fails to stabilize a conformation, and possibly the CaM explores new conformation, translating into a high RMSD (red line in Figure 7A). The CaM–5-HT complex remains stable during 100 ns of molecular dynamics, as observed in Figure 7B, where four fragments of the molecular dynamics trajectory were extracted, representing the structural models (initial, 25, 50, 75, and 100 ns).

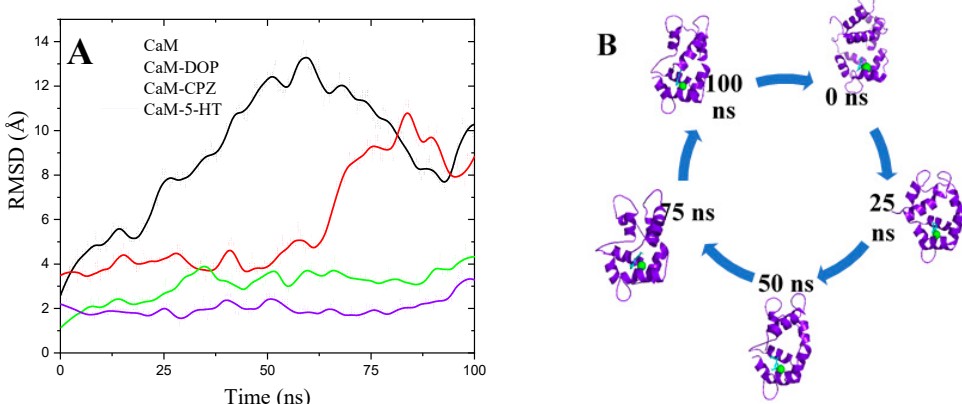

**Figure 7.** Analysis of 100 ns of molecular dynamics simulation. (**A**) The RMSD vs. time plot for 100 ns molecular dynamics simulation. (**B**) Structural models of fragments of the molecular dynamics of the CaM–5-HT complex.

## 4. Conclusions

The high sensitivity and specificity of the *h*CaM-M124C-*mBBr* fluorescent biosensor allow the detection of most anti-CaM ligands and endogenous metabolites such as 5-HT. The biosensor was designed based on experimental and theoretical data to place the fluorophore in the optimal position of the protein. The importance of reporting the direct interaction of 5-HT with CaM lies in proposing modulations of different proteins with which CaM interacts, among which are the GPCRs. Additionally, the binding of 5-HT to CaM may be related to some side effects of anti-CaM drugs such as CPZ, TFP, FLU, and VBT. On the other hand, the theoretical studies help us to complement experimental data and, in the case of the CaM–5-HT complex, to propose a structural model of it (docking), as well as its stability as a function of time (simulation of molecular dynamics).

**Supplementary Materials:** The following are available online at https://www.mdpi.com/article/10.3390/chemosensors9090250/s1, Figure S1: Chromatogram of the hydrophobic exchange purification of the CaM M124C protein, Figure S2: Chromatogram of the CaM M124C-*mBBr* protein on a size exclusion column (superdex75), Figure S3: Fluorescence spectra of *h*CaM M124C-*mBBr* adding known concentrations of DOP and AMA, Figure S4: Progress of the molecular dynamics simulation of the 4Ca2+-CaM–CPZ complex. The total energy of the system vs. time. The steps of the simulation comprise 50 ps of heating, 50 ps of equilibration at constant volume, 500 ps of equilibrium, and 100 ns of molecular dynamics simulation.

**Author Contributions:** Conceptualization, M.G.-A., L.X.V.-B., R.M., and A.S.-P.; methodology, M.G.-A., L.X.V.-B., I.V.-L., and P.C.-S.; software, M.G.-A., A.S.-P.; validation, M.G.-A., L.X.V.-B., I.V.-L., and P.C.-S.; formal analysis, M.G.-A., R.M., A.S.-P.; investigation, M.G.-A., L.X.V.-B., I.V.-L., P.C.-S., R.M., and A.S.-P.; resources, M.G.-A., R.M., and A.S.-P.; data curation, L.X.V.-B., I.V.-L., and P.C.-S.; writing—original draft preparation, M.G.-A., and L.X.V.-B..; writing—review and editing, M.G.-A., L.X.V.-B., I.V.-L., P.C.-S., R.M., and A.S.-P.; visualization, M.G.-A., L.X.V.-B., and I.V.-L.; supervision, M.G.-A.; project administration, M.G.-A., R.M., and A.S.-P.; funding acquisition, M.G.-A., R.M., and A.S.-P. All authors have read and agreed to the published version of the manuscript.

**Funding:** This work was supported by UNAM-PAPIIT IN203719 and DGTIC-UNAM (LANCAD-UNAM-DGTIC-313).

**Institutional Review Board Statement:** Not applicable.

**Informed Consent Statement:** Not applicable.

**Data Availability Statement:** Not applicable.

**Acknowledgments:** We are indebted to Dirección General de Cómputo y de Tecnologías de Información y Comunicación, UNAM, for providing the resources to carry out computational calculations through Miztli System.

**Conflicts of Interest:** The authors have declared that there is no conflict of interest.

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
