# Peer review of "Application of a Fluorescent Biosensor in Determining the Binding of 5-HT to Calmodulin"

_chemosensors, doi:10.3390/chemosensors9090250_

Round 1
Reviewer 1 Report
In this manuscript by Vasquez-Bochm and coworkers the application of a bioconjugate of the calcium-binding protein calmodulin (CaM) to serve as a sensor for measuring serotonin (5-HT) and chlorpromazine CPZ binding to the modified protein using fluorescence spectroscopy is presented. The main issue with this manuscript is that the authors provide very little details about the nature of the CaM-bioconjugate (hCaM-M124C-mBBr) used for these studies. Its synthesis was reported by some members of this group in reference 33 (of this manuscript) that was published in Anal. Biochem. 2009, 387, 64-70; 12-years ago! This same reference is also listed as # 18 in the current manuscript. From ref. 18/33 it appears that Met124 of CaM was replaced with a Cys124 residue and the thiol group of Cys124 as reacted with monobromobimane (mBBr) to afford the fluoresent bioconjugate of CaM. However, even in ref. 18/33 the bioconjugate was poorly characterized, with no mass spectrometry (MS) to acquire exact mass of the conjugate and MS-MS experiments to determine the site of modification. Figure 1A in the present manuscript is of poor quality, making it difficult for the reader to figure out how the hCaM-M124C-mBBr biosensor was created without going back to the original reference in 18/33. For publication in Chemosensors, the authors should provide more details on the nature of the CaM-bionjugate with accompanying MS results and chromatography in the Supporting Information to establish identity and purity of the bioconjugate.
Other issues: 1) why is the fluoresence of the mBBr label quenched upon addition of HT and CPZ (Figure 3)? 2) Figure 2 is not clear, difficult to read amino acid sequences. Also, why does the Figure highlight binding by trifluoperazine (TFP) when TFP was not used in the current binding studies?
Reviewer 2 Report
The concept has already been developed previously by the authors for high throughput screening of CaM inhibitors. In this paper, the authors developed a fluorescent biosensor to detect 5-HT. There are some important questions can be addressed in the manuscript:
- The authors claim the hCaM-M124C-mBBr fluorescent biosensor has a higher sensitivity. However, the detection limit in this case is in micromolar range. Authors need to provide a comparative analysis to show the sensitivity of their sensor as compared to previously developed.
- Further, there is no data to prove the specificity of the sensor as claimed in the conclusions.
- Did authors developed a calibration curve and quantified unknown concentrations to prove the applicability of the sensor?
- There is no comparative data of the hCaM-M124C-mBBr fluorescent biosensor with standard technique to prove its accuracy. For example, CaM kinase II assay can be used to detect ligands and compared with the sensor performance.
- The sensor has only been tested in buffers; however, it is very important that the sensor must be tested in more complex media/cultures.
- Did authors test the sensor for non-specific binding? It is also very important to test selectivity of the developed biosensor.
- It is important to discuss other detection methods developed for 5HT detection and compare the sensitivity, selectivity, and specificity with the developed methods.
Reviewer 3 Report
The paper entitled "Application of a fluorescent biosensor in the determination of 5-HT binding to calmodulin" presents the usefulness of using the hCaM-M124C-mBBr biosensor to determine the affinity of serotonin 5HT. I think it is an interesting work, the authors combine fluorescence experiments with molecular modeling tools, which in my opinion is of great help to understand how biosensors work and help to design better and more robust biosensors.
I think this work could be published.
Author Response
Thanks for your comments
Round 2
Reviewer 1 Report
The authors have done a good job addressing the concerns raised in my initial assessment of the paper. This revised manuscript is much improved, and this reviewer is supportive of publication in Chemosensors.
Reviewer 2 Report
The authors have responded to all my questions.